# On the Development of a Technical Specification for the Use of Fine Recycled Aggregates from Construction and Demolition Waste in Concrete Production

**DOI:** 10.3390/ma13194228

**Published:** 2020-09-23

**Authors:** Miguel Bravo, António P. C. Duarte, Jorge de Brito, Luis Evangelista, Diogo Pedro

**Affiliations:** 1Civil Engineering Research and Innovation for Sustainability, Department of Civil Engineering, Barreiro School of Technology, Polytechnic Institute of Setúbal, Rua Américo da Silva Marinho, 2839-001 Lavradio, Portugal; miguelnbravo@tecnico.ulisboa.pt (M.B.); antonio.duarte@estbarreiro.ips.pt (A.P.C.D.); 2Civil Engineering Research and Innovation for Sustainability, Department of Civil Engineering, Architecture and Georresources, IST—Universidade de Lisboa, Av. Rovisco Pais, 1049-001 Lisbon, Portugal; 3Civil Engineering Research and Innovation for Sustainability, Instituto Superior de Engenharia de Lisboa, R. Conselheiro Emídio Navarro, 1950-062 Lisboa, Portugal; evangelista@dec.isel.ipl.pt; 4Civil Engineering Research and Innovation for Sustainability, Instituto Superior Técnico (IST)—Universidade de Lisboa, Av. Rovisco Pais, 1049-001 Lisbon, Portugal; diogo.pedro@tecnico.ulisboa.pt

**Keywords:** recycled aggregate concrete, construction and demolition waste, fine recycled aggregates, technical specification proposal, sustainability

## Abstract

The possibility of using recycled aggregates from construction and demolition waste (CDW) in concrete is rather widely agreed upon when it comes to the use of coarse recycled aggregates. However, this is not the case when fine recycled aggregates (FRA) are considered, as it is deemed that these seriously impair the behaviour of concrete. Hence, this work presents a technical specification proposal for the use of FRA from CDW in concrete, to attempt to fill this gap in legislation. The specification is based on a wide collection of experimental results, from which it is shown that for low incorporation ratios (up to 25%), the properties of concrete with FRA from CDW are comparable to those of a reference concrete. The intended international scope of the specification is ensured by the fact that FRA from CDW are typified by composition (percentage of concrete, masonry, glass, etc.) rather than by geographical origin or construction type. It is shown that, after typifying the FRA and assuming, as per design, the acceptable percentage losses (relative to a reference concrete) of mechanical, durability-related and long-term physical properties, if the maximum incorporation ratios proposed of each type of FRA are used, the variation of properties remains within the limits established.

## 1. Introduction

The construction sector is amongst the most environmental impacting in Europe, consuming high amounts of raw materials and energy. Naturally, this contributes to a high generation of construction and demolition waste (CDW), which is the largest fraction of waste produced in the European Union: about 900 million tonnes of the total 3000 million tonnes of waste generated yearly [1]. Recycling of CDW with its re-introduction in the construction sector has emerged and has been shown to be economically and environmentally viable [2,3,4,5], constituting a solution not only to the reduction in the volume of waste deposited into landfills, but also to the reduction in the extraction of raw materials. However, CDW are still mostly used as flexible road pavement sub-bases and rarely used as recycled aggregates (RA) for concrete production. This derives from the fact that RA from CDW have a much higher variability of composition and show worse properties than those of natural aggregates (NA). In general, RA tend to be more porous and to have more elongated shapes and higher roughness than those of NA [6]. This results in more porous recycled aggregate concrete, not only due to the higher porosity of the RA but also to the need for higher amounts of mixing water to maintain workability. As a consequence, recycled aggregate concrete displays lower mechanical and durability-related properties than natural aggregate concrete.

The mechanical properties (compressive and tensile strengths and Young’s modulus) of concrete with RA from mixed CDW [7,8,9,10], concrete [11,12,13], ceramics [14,15,16] or glass [17,18] are generally lower than those of natural aggregate concrete, especially when high replacement percentages (e.g., higher than 25%) are considered. Etxeberria et al. [12] and Medina et al. [16] reported, respectively, an invariance and an increase of up to 11% of compressive strengths for replacement percentages of 25% of NA with RA. Barra [19] and Oliveira et al. [11] reported that, in order for concrete with RA to have the same compressive strength as that of a NA (reference) concrete, the cement content would have to be increased by between 7% and 25%. However, Kou et al. [7] and Poon et al. [8] found that the differences, respectively, in Young’s modulus and compressive strength, between concrete with RA from CDW and its NA counterpart, tended to fade over time. Decreases of 40% in Young’s modulus and of 19% in compressive strength at 28 days were reported to decrease, respectively, to 28% and to 10% at 90 days, respectively, for concrete mixes with full replacement of coarse NA with coarse RA. With respect to durability-related properties, as stated, the use of RA from CDW generally leads to a higher porosity of the resulting concrete, thus increasing their water absorption [20,21,22,23], carbonation depth [22,23,24] and chloride-ion penetration [7,22,23]. Nevertheless, some authors also state that the pozzolanic activity promoted by the use of some RA may improve some of the durability-related properties [24,25]. Due to the generally inferior performance of concrete with RA from CDW compared to natural aggregate concrete, some authors investigated the possibility of mitigating the outcomes of using RA by the inclusion of steel [26] or polypropylene [27] fibers or mineral additions such as silica fume [28] or fly ash [29] in concrete mixes. It is shown that fibers are able to increase the compressive and tensile strengths and toughness of concretes with RA from CDW [26,27] and that the mineral additions contribute to a more compact material and enhance the interfacial transition zone (ITZ), thus improving both their mechanical and durability-related properties [28,29].

Although the properties of concrete with RA from CDW generally decrease, the decreases are higher with the use of fine aggregates than with that of coarse aggregates [10,13,20,23,30]. Given this fact, both experimental investigations and, consequently, the development of legislation on the use of RA from CDW have been mostly focused on the use of coarse aggregates, excluding or limiting the use of fine aggregates to non-structural concrete [31]. Regarding the developments of legislation on the use of RA, between 2007 and 2008, EN 13,242 (on the use of aggregates for unbound and hydraulically bound materials for civil engineering applications) and EN 12,620 (on the use of aggregates for concrete) started to include clauses explicitly addressing the use of RA. In Portugal, the National Laboratory of Civil Engineering published in 2006 (and revised in 2009) the LNEC E471 specification, which establishes recommendations and minimum requirements for the use of coarse RA from CDW in the production of hydraulic binder concrete, but fine RA were excluded from possible utilizations. In brief, the LNEC E471 specification establishes three types of aggregates, based on their compositions: two types are mainly based on concrete products’ waste, and the minimum quantity of this constituent is imposed, and the third one is based on a mix of concrete and masonry products’ wastes. The first two types of RA can be used in simple or reinforced concrete applications, whereas the third one can only be used in filling and lean concrete applications. Finally, the specification imposes maximum concrete strength classes on which the first two types of aggregates may be used.

Given the absences of legislation on the use of fine RA from CDW, this work aims at contributing to an increment of the use of these aggregates in concrete applications by the construction industry. This is made by presenting a proposal of specification that is in part inspired by LNEC E471 (for coarse RA), but founded on the development of an extensive experimental campaign on the study of concrete mixes with the use of fine RA from CDW. The specification proposal intends to have an international scope, this being ensured by the typifying of FRA from CDW by composition (percentage of concrete, masonry, glass, etc.) rather than by geographical origin or construction type. The main novelty of this work, aiming at advancing the in-depth understanding between the properties of RA from CDW and the resulting concrete produced, is two-fold: (i) presentation of a specification for the use of fine RA from CDW in concrete (still absent) and (ii) the development of rules accounting for RA composition (rather than origin) and based on an assumed decrease in properties (rather than on the imposition of minimum concrete strength classes). To this end, the main objectives of the work, which also coincide with the following sections, are:-A brief description of the experimental campaign conducted and of the results obtained;-A detailed presentation of the technical specification;-Validation of the specification considering the FRA used in the experimental campaign.

## 2. Summary of Experimental Campaign

In this section, a brief description of the experimental campaign that was used to support the development of the proposed technical specification is made. A detailed description of the experimental procedures and results can be found in [10,23].

A total of 13 concrete mixes were studied, including one reference concrete (RC) mix and 12 recycled aggregate concrete (RAC) mixes comprising RA from CDW from three recycling plants in Portugal (Vimajas, in Sintra, Europontal (Figure 1a), in Faro, and Ambilei, in Leiria). The RA were incorporated at four contents (10%, 25%, 50% and 100%) as replacement, in volume, of the fine natural aggregates. After determining the Faury’s curve for the RC mix composition, the aggregates in the concrete mixes with FRA from CDW were replaced sieve by sieve.

Concrete mixes were produced with 350 kg/m^3^ of Portland cement CEM I 42.5 R and with tap water. River sand and crushed limestone aggregates with maximum particle sizes of 4.0 mm and of 22.4 mm, respectively, were used. The target workability of all mixes was that corresponding to a slump class S3 (125 ± 15 mm slump measured with the Abrams cone) and the target strength class of the RC was C30/37. The mixing procedure was as follows: the RA were introduced into the mixer with 2/3 of the mixing water, to allow water absorption (4 min), the NA were added afterwards (2 min) and finally, the cement and remaining 1/3 of mixing water were introduced (4 min).

Recycled aggregates used were both physically and chemically characterized. The following tests were performed: particles’ density (EN 1097-6, 2013), water absorption (EN 1097-6, 2013), water soluble sulphate content (EN 1744-1, 2010), acid soluble sulphate content (EN 1744-1, 2010) and total sulphur content (EN 1744-1, 2010). The obtained results are shown ahead in Section 4. 

Regarding the characterization of concrete in the hardened state, the following properties were tested: (i) compressive strength in cubes and (ii) in cylinders at 28 days (NP EN 12390-3, 2011), (iii) splitting tensile strength at 28 days (EN 12390-6, 2009), (iv) Young’s modulus (LNEC E397, 1993), (v) water absorption by immersion (LNEC E394, 1993) and (vi) by capillarity (LNEC E393, 1993), (vii) resistance to carbonation at 28 days (LNEC E391, 1993), (viii) resistance to chloride-ion penetration at 28 days (LNEC E463, 2004), (ix) shrinkage at 91 days (LNEC E398, 1993) (Figure 1b) and (x) creep at 91 days (LNEC E399, 1993). Three specimens were used in each test, except for the determination of water absorption by immersion and of compressive strength in cubes in which, respectively, 4 and 5 specimens were used.

Table 1 shows the average value ± standard deviation values of the mechanical properties obtained, in particular: compressive strength in cubes (f_cm,cubes_), compressive strength in cylinders (f_cm,cylinders_), splitting tensile strength (f_ctm_) and Young’s modulus (E). Moreover, Table 2 shows the average ± standard deviation values of the durability-related and long-term physical properties, namely: water absorption by immersion (ΔW_im_), water absorption by capillarity after 72 h (ΔW_cap_), carbonation depth (Z), chloride-ion diffusivity coefficient (D), extension due to shrinkage (ε_sh_) and extension due to creep (ε_cr_). In both tables, the following labelling of the concrete mixes is adopted: “RC” stands for the reference concrete (with no FRA), as usual, and for the RAC mixes, taking “V10” as an example, the first letter stands for the FRA origin, “V” for Vimajas, “E” for Europontal and “A” for Ambilei, and the number after it for the percentage replacement of fine NA with fine RA from CDW (10%, 25%, 50% or 100%).

In general, it can be seen that the incorporation of fine RA leads to a decrease in the concrete properties and this decrease is generally higher as the replacement percentage increases, although for the lowest contents studied (10% and 25%) the properties’ variations are generally low and, at times, even some minor improvements in the properties can be achieved (this may also be confirmed ahead in Figure 2 and Figure 3). This is in agreement with that reported by Etxeberria et al. [12] and Medina et al. [16] and means that for low incorporation ratios of FRA, concrete mixes with properties comparable to those of a standard (reference) concrete can be obtained. It can also be concluded that RA from Vimajas led to the worst performances, RA from Ambilei to the best performances and RA from Europontal to intermediate results. This naturally stems from their composition, as can also be seen ahead in detail: Ambilei has the highest percentage of concrete products and Vimajas has the highest content of bituminous materials, clay soils and polymers. Nevertheless, the reduction in the mechanical properties can be mainly explained by the lower strength and stiffness of RA compared to natural aggregates. Naturally, not only strength and stiffness but also the failure modes of the concrete mixes are important to characterize their mechanical behaviour. Given the broad scope (and number of tests) of the experimental campaign that served as the foundation of the current work, for instance, the brittleness/ductility of the mixes was not explicitly assessed. However, regarding the cracking patterns of the mixes, it was observed that in concretes with 100% RA from CDW, the cracking surface, obtained with the splitting tensile strength test, was rather irregular, confirming the existence of intact coarse aggregates along the cracked surface. This means that failure occurred mainly by the ITZ between the FRA and the cement paste (Figure 1c). The weaker ITZ of these concrete mixes stemmed from the higher water/cement ratio required to maintain the set workability, given the lower density and higher roughness of FRA from CDW. With respect to the durability-related properties, it can mainly be concluded that concrete with FRA has higher porosity than its reference counterpart. This is mainly explained by the higher porosity of some of the FRA and the higher porosity of the cement paste, given the higher water required to maintain the workability used in the production of the RAC. Finally, the decay of the long-term physical properties (shrinkage and creep) can be mainly explained by the lower stiffness of FRA and the higher porosity of RAC mixes that ultimately decrease their stiffness: the “softer” RA do not oppose the cement paste shrinkage as it loses excess water (not required for cement hydration) and the “softer” paste (and aggregates) are more deformable under an applied stress than those of a standard (reference) concrete.

## 3. Proposal of a Specification

### 3.1. Scope

The present specification proposal aims at providing recommendations and establishing minimum requirements for the use of FRA in the production of hydraulic binder concrete. As stated, the proposed specification stems from an in-depth assessment of the consequences of the incorporation in concrete of fine RA from CDW sourced from the three recycling plants presented (Ambilei, Europontal and Vimajas).

This specification proposal follows the existing specification (E 471), which establishes the conditions for the use of coarse RA in the production of hydraulic binder concrete, and intends to overcome the lack of regulation on the use of fine RA in concrete.

### 3.2. Framework

CDW were identified by the European Commission as a waste stream whose treatment and recycling are a priority, given the high quantity generated and its high potential for reuse as a raw material. Proper management and recycling of this type of waste leads to a reduction (i) in the consumption of natural resources and (ii) in the amount of landfilled waste, both measures with beneficial effects on the environment.

In view of this objective, Community Directive 2008/98/EC establishes that the European Union member states must take the required measures to achieve a minimum of 70% reuse of the CDW produced by 2020 [32].

In order to fulfil this objective, the Portuguese Government published Decree No. 46/2008 of March 12, which establishes the regime for CDW management operations, covering their prevention and reuse and their collection operations, transport, storage, sorting, treatment and recovery. This document encourages the separation by type of CDW in the construction site and its local subsequent reuse, thus mitigating the use of energy for transportation to a different location. It is mandatory to separate non-inert materials which should be made “by streams and rows of materials, for recycling or other forms of recovery”. The separation of all these constituents, when not carried out immediately on site, must be done at a CDW recycling plant. This decree is also intended to mitigate the disposal of waste in illegal landfills.

Recycled CDW is essentially divided into three parts: the inert or mineral part, such as concrete, brick, mortar, sand and stone; the non-inert part, such as metals, plastics, glass, cardboard and paper; and the hazardous waste, which must be recycled through specifically prepared means.

Since 2004, the European List of Waste (ELW) has been applied in Portugal, published in Diário da República through Decree-Law No. 209/2004. The ELW ensures the harmonization of existing regulations regarding the identification and classification of waste, allowing separation, transport, and recovery operations to be carried out in a more appropriate manner. Using a six-digit code in the list, it is possible to identify the origin of the waste, as well as the specific type of waste. CDW are identified with code 17—Construction and demolition waste (including soil excavated from contaminated sites). The remaining four digits of the code identify the type of waste such as concrete (e.g., 17 01 01) or wood (e.g., 17 02 01), amongst others. The recycling plants sort the separated material, which may or not be inert, identifying it with the respective code, and send the data to the Portuguese Environment Agency, which is in charge of establishing statistical data.

With respect to the use of aggregates from CDW in concrete, it is agreed that the replacement of part of the coarse NA with coarse RA does not significantly affect the properties of concrete. However, this is not the case when fine NA are replaced with fine RA. Therefore, the fine RA, having an expected worse behaviour compared to their coarse counterparts, were therefore initially discarded from experimental investigations. This led to a near-absence of legislation regarding their use.

### 3.3. Aggregates Classification

This specification proposal begins with the definition of three types of aggregates from CDW. These three types of RA have different requirements for their composition (Table 3).

Subsequently, in this specification proposal, four classes of RA utilization are defined. Each class corresponds to different quantitative decreases in the different properties of concrete, compared to a reference concrete (standard concrete). This approach is distinct from that adopted by LNEC E471 Specification as it does not limit the strength classes on which fine RA can be used. For instance, in Table 4, Class 1 comprises the incorporation of one of the three types of FRA in percentages (shown in Table 5) that result in a decrease in the mechanical performance of the resulting concrete not exceeding 15% and of the remaining (durability-related and long-term physical) properties not exceeding 35%. The variation values corresponding to classes 2, 3 and 4 are also presented in Table 4.

The allowed percentages of incorporation of each of the three types of fine RA in order to comply with the quantitative requirements imposed for each class are shown in Table 5. For this purpose, the results obtained in some of the tests carried out over the vast experimental campaign conducted, and described previously, were used. The analysed test results were (i) the compressive strength in cubes and (ii) in cylinders at 28 days, (iii) the splitting tensile strength at 28 days, (iv) the Young’s modulus, (v) the water absorption by immersion and (vi) by capillarity, (vii) the resistance to carbonation at 28 days, (viii) the resistance to chloride-ion penetration at 28 days, (ix) shrinkage at 91 days and (x) creep at 91 days.

### 3.4. Properties and Minimum Requirements

The fine RA, within any of the three defined types, must comply with the requirements established in this specification, with respect to a given set of physical properties as shown in Table 6.

### 3.5. Implementing Rules

The use of the three types of fine RA defined in the production of hydraulic binder concrete is subjected to the conditions set out in this specification proposal. The use of fine RA of different composition requires further specific studies to assess their influence on the fundamental properties of concrete.

As seen in Table 4 and Table 5, the incorporation ratios of one of the three types of FRA included in class 1 should be used only when the production of a concrete with a decrease in mechanical properties lower than 15% and a decrease in the remaining properties lower than 35%, compared to a standard (reference) concrete, is acceptable. The fine RA incorporation ratios included in class 2 should only be made when the production of concrete is acceptable with a decrease in mechanical performance below 25% and a decrease in the remaining properties below 50%, and so on. However, it should be stressed that the contents of fine RA within class 4 are only for filling and regularization concrete applications.

### 3.6. Quality Control

The minimum conformity requirements for fine RA for all applications, shown in Table 6, should be checked by the producer with the minimum frequencies indicated in Table 7, since the composition of RA from CDW has a high variability.

The use of fine RA with compositions other than those shown in Table 3 requires further specific studies. Hence, it is essential that a weekly analysis on the constitution of fine RA is performed.

## 4. Validation of the Proposed Specification

As mentioned, the specification proposal presented in the previous sub-section stemmed from an in-depth assessment of the implications of the incorporation in concrete of fine RA from CDW from three recycling plants (Ambilei, Europontal and Vimajas).

In this section, the use of the specification proposal is exemplified and validated using the FRA from CDW studied. The main goals are to show (i) how the specification proposal should be used and (ii) that if the maximum percentage of incorporation of FRA from CDW defined in Table 5 for each type of FRA is adopted, the decreases in properties established in Table 4 can be fulfilled.

Hence, the first step is to determine the type of FRA from CDW (I, II or II), according to Table 3, taking into account their composition. Table 8 presents the results obtained from the composition determination of the FRA from CDW.

Ambilei’s FRA are classified as type I since they are constituted, at more than 80%, by “concrete, concrete products, mortar, non-bonded aggregates, natural stone and aggregates treated with hydraulic binders”. Both Europontal’s and Vimajas’ FRA have contents of the former lower than 80%, but the sum of that fraction and of masonry is higher than 80%. In particular, for Europontal’s RA, that sum is higher than 90% and the sum of remaining components is lower than 10%, thus making these aggregates of type II. Finally, since the sum of the remaining components of Vimajas’ RA is higher than 10% (though lower than 20%), these aggregates are of type III.

After classing the RA, there are still minimum properties, e.g., density, water absorption and sulphur-related contents, that must be guaranteed, according to Table 6, and that should be determined by producers with the frequencies determined in Table 7. Table 9 shows the properties of the analysed aggregates, allowing the conclusion that all three types are in conformity with the proposed requirements.

Taking into account the results obtained in the experimental campaign developed (summarized in Section 2), Figure 2 and Figure 3 show the percentage variations (of the average values) respectively, of the mechanical and of the durability-related and long-term physical properties of the concrete mixes with the different fine RA with respect to the RC. Additionally, in Figure 2 and Figure 3, the concrete classes (from 1 to 4) defined in Table 4 are also shown in different colours as well as their boundaries, in dashed lines. The percentage variations were determined with the values presented in Table 1 and Table 2. It should be emphasized that if a given mechanical property of a concrete with fine RA is lower than that of the reference concrete, the variation is naturally shown as being negative in Figure 2. However, in the case of durability-related and long-term physical properties, the variation is considered to be negative if there is an increase in a coefficient associated with a given property, since it is more detrimental to the concrete performance: e.g., increase in carbonation depth or increase in shrinkage.

Finally, Table 10 summarizes the validation of the maximum incorporation ratios of the fine RA, showing that they result in the assumed decreases in mechanical and durability-related and long-term physical properties.

As Figure 2 and Figure 3 and Table 10 show, it seems valid to assume that for class 1 of concrete up to 25%, 10% and 5% of Type I, II, and II FRA can be used, respectively. These maximum ratios increase to 50%, 25% and 10%, for class 2 concrete and are even greater for class 3 concrete, allowing for 100%, 50% and 25% replacement ratios, for each of the established FRA types.

## 5. Conclusions

In this work, a specification proposal for the use of fine recycled aggregates, FRA, from construction and demolition waste, CDW, in concrete was presented. The specification proposal was inspired on the philosophy of the existing one on the use of coarse recycled aggregates, CRA, in Portugal-LNEC E 471, but modified in order for FRA to be used. Firstly, a brief literature review on concrete with CDW and on the existing standards on the use of recycled aggregates was presented. Then, the experimental campaign on which the specification proposal was founded was described and its results presented. Finally, the specification proposal was presented and its validation was made with the use of the FRA employed in the experimental campaign. The following main conclusions can be drawn:-Existing literature shows that the use of FRA from CDW in concrete is feasible but legislation on their use is still nearly absent (but needed). This is due to the fact that FRA lead to concrete with lower properties than those where coarse recycled aggregates are used;-Decreases in (i) mechanical (strength and stiffness), (ii) durability-related (water absorption, carbonation and chloride-ion penetration) and (iii) long-term physical (shrinkage and creep) properties of concrete with FRA were generally obtained experimentally with the increase in the incorporation ratio of FRA from CDW in concrete. The decreases in properties are mainly due to the higher porosity and lower strength and stiffness of FRA compared to fine natural aggregates;-For low incorporation ratios of FRA from CDW, of up to 25%, concrete mixes exhibit properties that are comparable to those of the reference concrete. For incorporation ratios higher than 25%, decreases of between 2% and 5%, between 6% and 17% and between 4% and 19% per each 10% increase in FRA percentage were obtained, respectively, for mechanical, durability-related and long-term physical properties;-The proposed technical specification was successfully validated since, when the maximum percentages of each type of FRA proposed in the specification were considered, the variation of properties remained within the limits established for a given concrete class. For instance, the use of (i) 25% FRA with more than 80% concrete or mortar content, or (ii) 10% FRA with more than 90% concrete, mortar and masonry combined, or (iii) 5% FRA with more than 80% concrete, mortar and masonry combined, resulted in concretes with decreases in the mechanical and remaining (durability-related and long-term physical) properties, respectively, lower than 15% and 35%;-The proposed technical specification is intended for an international scope due to the fact that the FRA are typified based on their composition rather than on their geographical origin or construction type. This may contribute to an increase in the acceptance by the construction industry of the use of FRA from CDW in concrete applications.

## Figures and Tables

**Figure 1 materials-13-04228-f001:**
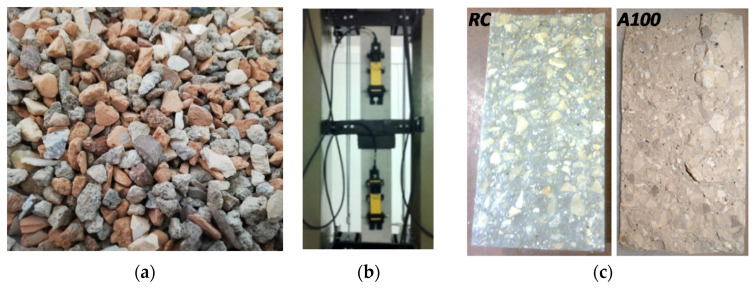
Details of (**a**) FRA from Europontal recycling plant, (**b**) shrinkage test and (**c**) cracking patterns of RC and A100 after the splitting tensile strength test.

**Figure 2 materials-13-04228-f002:**
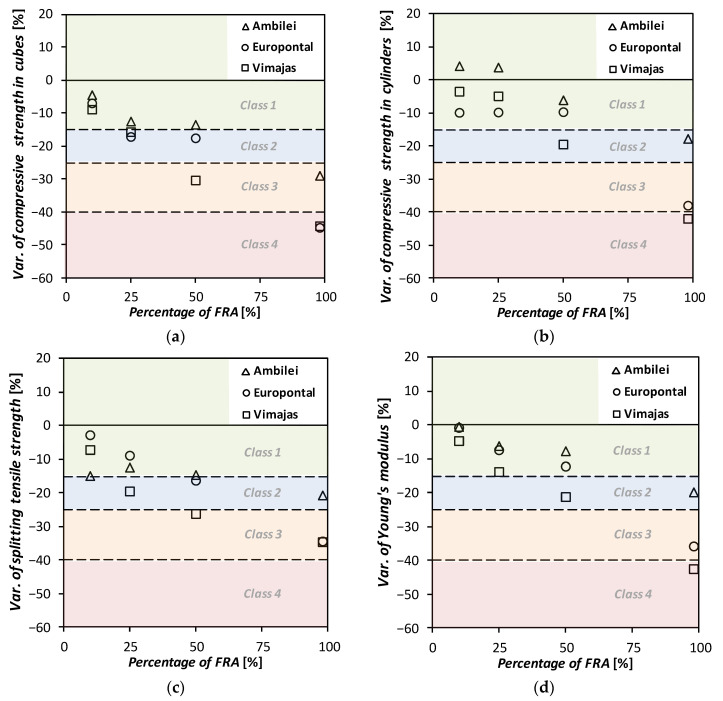
Variation of compressive strength in (**a**) cubes and (**b**) cylinders, (**c**) of splitting tensile strength and (**d**) of Young’s modulus of concrete made with FRA, relative to the reference concrete.

**Figure 3 materials-13-04228-f003:**
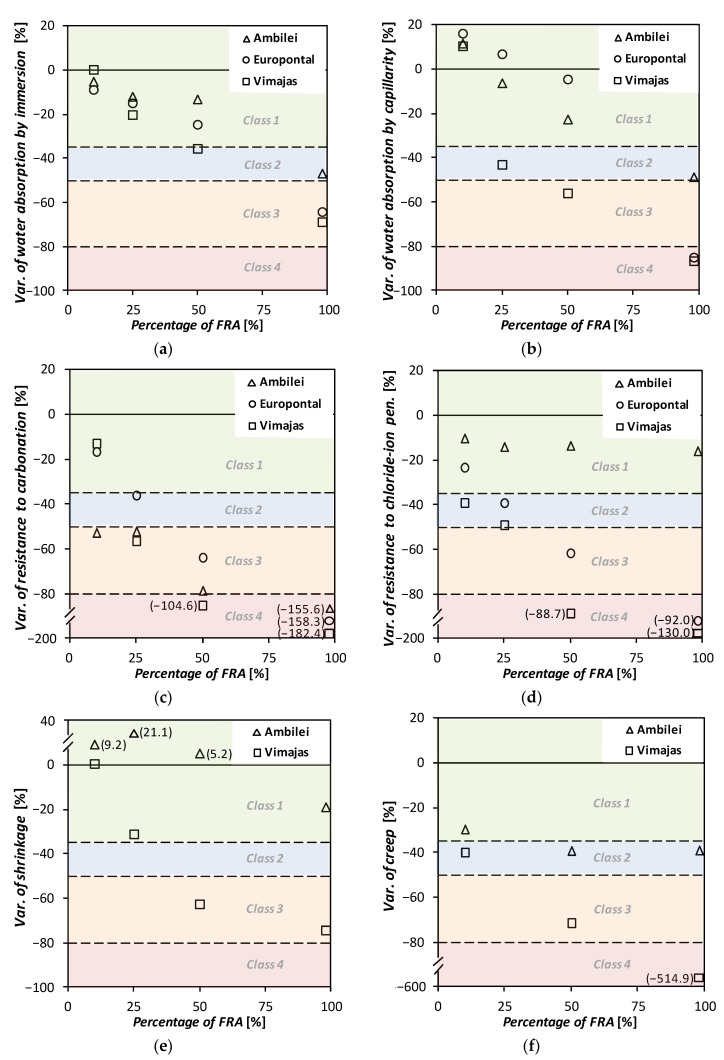
Variation of water absorption (**a**) by immersion and (**b**) by capillarity, of resistance to (**c**) carbonation and (**d**) chloride-ion penetration, (**e**) of shrinkage and (**f**) of creep of concrete made with FRA, relative to the reference concrete.

**Table 1 materials-13-04228-t001:** Mechanical properties of the concrete mixes.

Concrete Mix	f_cm,cubes_ (MPa)	f_cm,cylinders_ (MPa)	f_ctm_ (MPa)	E (GPa)
RC	53.9 ± 1.8	37.5 ± 1.3	4.0 ± 0.0	40.5 ± 0.2
V10	49.2 ± 1.1	36.2 ± 0.9	3.7 ± 0.1	38.6 ± 0.9
V25	45.6 ± 1.5	35.7 ± 0.1	3.2 ± 0.2	34.9 ± 0.5
V50	37.6 ± 1.3	30.2 ± 1.3	3.0 ± 0.2	31.9 ± 0.2
V100	30.2 ± 0.5	21.8 ± 1.8	2.6 ± 0.0	23.3 ± 0.6
E10	50.3 ± 1.5	33.8 ± 3.5	3.9 ± 0.2	40.2 ± 0.0
E25	44.7 ± 2.0	33.8 ± 2.5	3.7 ± 0.2	37.5 ± 0.3
E50	44.5 ± 1.0	33.9 ± 3.1	3.4 ± 0.5	35.6 ± 0.2
E100	29.9 ± 0.6	23.3 ± 0.1	2.6 ± 0.1	26.0 ± 0.6
A10	51.6 ± 1.0	39.1 ± 3.7	3.4 ± 0.1	40.3 ± 0.3
A25	47.3 ± 1.1	38.9 ± 1.4	3.5 ± 0.2	38.0 ± 0.2
A50	46.8 ± 1.2	35.2 ± 0.4	3.4 ± 0.3	37.4 ± 0.4
A100	38.4 ± 1.2	30.8 ± 0.9	3.2 ± 0.1	32.5 ± 0.6

**Table 2 materials-13-04228-t002:** Durability-related and long-term physical properties of the concrete mixes.

Concrete Mix	ΔW_im_ [%]	ΔW_cap_ (× 10^−3^ g/mm^2^)	Z (mm)	D (× 10^−12^ m^2^/s)	ε_sh_ (µstrain)	ε_cr_ (µstrain)
RC	12.9 ± 0.3	2.14 ± 0.14	4.5 ± 0.4	12.7 ± 1.3	−306.0 ± 7.6	−466.0 ± 50.0
V10	12.9 ± 0.2	1.91 ± 0.05	5.1 ± 0.4	17.6 ± 1.1	−304.0 ± 9.2	−682.0 ± 29.2
V25	15.5 ± 0.3	3.06 ± 0.14	7.0 ± 0.4	18.9 ± 0.8	−401.0 ± 18.8	- *
V50	17.5 ± 0.5	3.34 ± 0.05	9.2 ± 0.8	23.9 ± 0.7	−498.0 ± 17.5	−765.0 ± 0.0
V100	21.8 ± 0.2	3.99 ± 0.09	12.7 ± 0.9	29.2 ± 1.9	−533.0 ± 104	−3270.0 ± 210.0
E10	14.0 ± 0.6	1.79 ± 0.06	5.3 ± 0.4	15.6 ± 0.7	- *	- *
E25	14.8 ± 0.2	1.99 ± 0.12	6.1 ± 0.2	17.7 ± 0.6	-	-
E50	16.1 ± 0.4	2.24 ± 0.29	7.4 ± 1.0	20.5 ± 0.6	-	-
E100	21.2 ± 0.2	3.96 ± 0.23	11.6 ± 0.4	24.3 ± 0.5	-	-
A10	13.6 ± 0.3	1.89 ± 0.04	6.9 ± 0.4	14.0 ± 1.7	−278.0 ± 0.0	−612 ± 115
A25	14.4 ± 0.4	2.27 ± 0.05	6.9 ± 0.1	14.5 ± 1.2	−241.0 ± 11.3	- *
A50	14.6 ± 0.5	2.62 ± 0.03	8.0 ± 0.4	14.4 ± 0.7	−290.0 ± 0.0	−682.0 ± 150.0
A100	18.9 ± 0.4	3.18 ± 0.27	11.5 ± 0.3	14.7 ± 0.4	−364.0 ± 6.3	−647.0 ± 73.8

* Due to logistical problems, these tests could not be performed.

**Table 3 materials-13-04228-t003:** Composition of the three types of fine RA.

Composition (in Percentage of Mass)	Fine RA Type I	Fine RA Type II	Fine RA Type III
Concrete, concrete products, mortar, non-bonded aggregates, natural stone and aggregates treated with hydraulic binders	≥80	≥90	≥80
Masonry	≤20		
Bituminous materials		≤10	≤20
Glass			
Other materials ^1^			

^1^ Clay soils, polymers (plastics and rubbers), metals, non-floating wood, and plaster.

**Table 4 materials-13-04228-t004:** Classification of FRA as a function of the consequences of their incorporation in concrete properties.

Classification	Variation of Properties with Respect to Standard Concrete	
	Decrease of Mechanical Properties	Decrease of Durability-Related and Long-Term Physical Properties
Class 1	≤15%	≤35%
Class 2	≤25%	≤50%
Class 3	≤40%	≤80%
Class 4 (non-structural concrete)	>40%	>80%

**Table 5 materials-13-04228-t005:** Maximum incorporation ratios of FRA for each class.

Classification	Maximum Incorporation of FRA
Class 1	25% RA Type I
	10% RA Type II
	5% RA Type III
Class 2	50% RA Type I
	25% RA Type II
	10% RA Type III
Class 3	100% RA Type I
	50% RA Type II
	25% RA Type III
Class 4 (non-structural concrete)	>50% RA Type II
	>25% RA Type III

**Table 6 materials-13-04228-t006:** Properties and minimum conformity requirements of FRA for all applications.

Properties	Test Standard	Conformity Requirements
Particles density (kg/m^3^)	EN 1097-6	≥2000
Water absorption (%)	EN 1097-6	≤14
Water soluble sulphate content (%)	EN 1744-1	≤0.2
Acid soluble sulphate content (%)	EN 1744-1	≤0.8
Total sulphur content (%)	EN 1744-1	≤1.0

**Table 7 materials-13-04228-t007:** Minimum composition determination frequencies.

Properties	Frequency
Particles’ density	Once a week
Water absorption	Once a week
Water soluble sulphate content	Once a week
Acid soluble sulphate content	Twice a year
Total sulphur content	Twice a year

**Table 8 materials-13-04228-t008:** Composition of the FRA analysed.

Composition (in Percentage of Mass)	Specification Proposal			Analysed FRA		
	Fine RA Type I	Fine RA Type II	Fine RA Type III	Ambilei	Europontal	Vimajas
Concrete, concrete products, mortar, non-bonded aggregates, natural stone and aggregates treated with hydraulic binders	≥80	≥90	≥80	83.7	68.8	75.2
Masonry	≤20			0.9	26.5	11.6
Bituminous materials		≤10	≤20	0.0	1.0	10.5
Glass				15.4	3.4	1.0
Other materials				0.0	0.3	1.7

**Table 9 materials-13-04228-t009:** Properties and minimum conformity requirements of the FRA established in the specification proposal.

Properties	Conformity Requirements	Analysed FRA		
		Ambilei	Europontal	Vimajas
Particles’ density (kg/m^3^)	≥2000	2112	2070	2063
Water absorption (%)	≤14	12.9	10.1	10.4
Water soluble sulphate content (%)	≤0.2	0.11	0.18	0.04
Acid soluble sulphate content (%)	≤0.8	0.2	0.8	0.1
Total sulphur content (%)	≤1.0	0.1	0.3	<0.1

**Table 10 materials-13-04228-t010:** Validation of the maximum incorporation ratios of fine RA.

	Class 1—DECREASE in Mechanical Properties Up to 15% and Decrease in Durability-Related and Long-Term Physical Properties Up to 35%			Class 2—Decrease in Mechanical Properties Up to 25% and Decrease in Durability-Related and Long-Term Physical Properties Up to 50%			Class 3—Decrease in Mechanical Properties Up to 40% and Decrease in Durability-Related and Long-Term Physical Properties Up to 80%		
Variation of Property with Respect to RC	25% of FRA from Ambilei	10% of FRA from Europontal	5% of FRA from Vimajas	50% of FRA from Ambilei	25% of FRA from Europontal	10% of FRA from Vimajas	100% of FRA from Ambilei	50% of FRA from Europontal	25% of FRA from Vimajas
f_cm,cubes_	−12.5	−6.8	−4.4 *^1^	−13.3	−17.0	−8.7	−28.8	−17.4	−15.5
f_cm,cylinders_	+3.8	−9.9	−1.8 *^1^	−6.1	−9.8	−3.5	−17.8	−9.7	−4.9
f_ctm_	−12.4	−2.8	−3.6 *^1^	−14.6	−8.9	−7.2	−20.7	−16.3	−19.5
E	−6.2	−0.8	−2.4 *^1^	−7.7	−7.4	−4.7	−19.8	−12.2	−13.8
ΔW_im_	−12.0	−8.9	+0.1 *^1^	−13.2	−15.0	+0.2	−46.9	−24.7	−20.3
ΔW_cap_	−6.3	+16.1	+5.2 *^1^	−22.7	+6.8	+10.4	−48.8	−4.6	−43.2
Z	- *^4^	−16.7	−6.5 *^1^	- *^4^	−36.1	−13.0	*^4^	−63.9	−56.5
D	−14.1	−23.3	−19.6 *^1^	−13.6	−39.2	−39.1	−16.0	−61.7	−49.1
ε_sh_	−21.1	- *^3^	−0.3 *^1^	−5.2	*^3^	−0.5	−19.0	*^3^	−31.2
ε_cr_	−33.3 *^2^	- *^3^	−20.0 *^1^	−39.3	*^3^	−40.0	−39.1	*^3^	−51.8 *^2^

*^1^ Obtained by linear interpolation of the results for 10% FRA from Vimajas. *^2^ Obtained by linear interpolation of the results for concrete with 25% and 50% of FRA. *^3^ Concrete with FRA from Europontal were not tested, as referred, for shrinkage and creep. *^4^ These results were considered abnormal.

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
