# Peer review of "On the Development of a Technical Specification for the Use of Fine Recycled Aggregates from Construction and Demolition Waste in Concrete Production"

_materials, 2020, doi:10.3390/ma13194228_

Round 1

Reviewer 1 Report

As high-quality fine aggregates that can be used in concrete manufacturing are being depleted, study on the proposal of technical specifications for fine recycled aggregates can be considered a timely topic. To propose the specification, in this study extensive experimental work and a review of existing standards were carried out. However, the reviewer is concerned that the experimental data and results are not sufficient to suggest such a specification. This can be overcome by changing the title to better fit the scope and result of the study. Therefore, the reviewer advises the authors to change the title more conservatively.

In addition, the abstract written is not suitable for a type of research article; it has to be rewritten. A significant portion of the abstract before “In this investigation~” (line 20-67) is written like an introduction; shorten this part to less than 3 sentences. Also, improve the abstract to clearly show the purpose of this study, the important findings or results, and the method to derive the results.

In the case of the current manuscript, the introduction section can consist of 4 paragraphs: the first (line 45-61), second (line 62-78), third (line 79-96), and the last paragraphs (line 97-104).

The conclusion section also needs to be improved. Revise it to clearly show the main findings the study and the authors' conclusions.

Author Response

Reviewer #1 COMMENTS/CORRECTIONS:
Comment 1: As high-quality fine aggregates that can be used in concrete manufacturing are being depleted, study on the proposal of technical specifications for fine recycled aggregates can be considered a timely topic. To propose the specification, in this study extensive experimental work and a review of existing standards were carried out.

Reply: The authors gratefully acknowledge the general comment made by Reviewer #1.

Action: The authors considered the suggestions made in the following comments.

Comment 2: However, the reviewer is concerned that the experimental data and results are not sufficient to suggest such a specification. This can be overcome by changing the title to better fit the scope and result of the study. Therefore, the reviewer advises the authors to change the title more conservatively.

Reply: The authors agree with the comment made by the Reviewer in the sense that, as it is, the title suggests that the topic is closed. Indeed, this is a first presentation of specification proposal but a final version should be based on a wider collection of results.

Action: The title was modified in accordance with Reviewer #1’s comment.

Comment 3: In addition, the abstract written is not suitable for a type of research article; it has to be rewritten. A significant portion of the abstract before “In this investigation~” (line 20-67) is written like an introduction; shorten this part to less than 3 sentences. Also, improve the abstract to clearly show the purpose of this study, the important findings or results, and the method to derive the results.

Reply: The authors agree with Reviewer #1 as the abstract was not clearly responding to the following points: scope, main objective/motivation, methods and main findings.

Action: The abstract was rewritten accordingly.

Comment 4: In the case of the current manuscript, the introduction section can consist of 4 paragraphs: the first (line 45-61), second (line 62-78), third (line 79-96), and the last paragraphs (line 97-104).

Reply: The authors thank the Reviewer for this comment. In fact, too many paragraphs were presented, imposing unneeded breaks in the text.

Action: The authors reformatted the ‘Introduction’ so it now consists of the 4 logically-organized paragraphs proposed by Reviewer #1.

Comment 5: The conclusion section also needs to be improved. Revise it to clearly show the main findings the study and the authors' conclusions.

Reply: The authors agree with this comment made by Reviewer #1. The conclusions are too narrative and do not concisely show the main findings and conclusions of the study.

Action: The authors modified and reformatted the conclusions accordingly, now in points/bullets.

Reviewer 2 Report

The reviewer feels that this manuscript is well written and organized to present the specifications of using FRA in concrete mixing. 

I do not feel that the revision is required and hence acceptable as it is in the present form.

Author Response

Reviewer #2 COMMENTS/CORRECTIONS:

Comment 1: The reviewer feels that this manuscript is well written and organized to present the specifications of using FRA in concrete mixing. I do not feel that the revision is required and hence acceptable as it is in the present form.

Reply: The authors gratefully acknowledge the general comment made by Reviewer #2.

Action: According to the comment made by Reviewer #2, no actions are required.

Reviewer 3 Report

The submitted manuscript presents a proposal for a technical specification for using fine recycled aggregates (RA) from construction and demolition waste (CDW) in concrete production. For this, several experimental results on recycled concrete (RC) obtained by the authors and produced using fine RA from three recycling plants in Portugal, with different contents of fine RA as replacement of fine natural aggregates, were analysed. The structure of the proposed technical specification was inspired from an existing Portuguese technical specification for using coarse RA from CDW in concrete production, in which fine RA were excluded.

The manuscript summarizes the experimental campaign and discusses the results. It Is shown that for contents of fine RA not to high, the decrease on the properties and durability of the obtained RC are not excessively reduced. The proposal for the technical specification is presented, in order to provide recommendations and establish minimum requirements for the use of fine RA in the production of RC. Finally, the use of the proposed technical specification is exemplified for RC using fine RA from the CDW studied in the experimental campaign.

The topic that is developed in the study is interesting and important since it is related with the sustainability of construction sector, and deals with fine RA in concrete for which existent legislation is still practically inexistent. The proposed technical specification could help task groups to propose future recommendations and also industries from the construction sector.

I made some suggestions in order to improve the manuscript. The authors should take the suggestions into account and revise their manuscript.

Comment 1

Section 1

In the literature review, the number of cited articles from at least one of the authors (self-citations) is to high (about 50%). Given the high number of published studies on the topic that can be found in the literature, this issue should be revised by reducing the number of self-citations and/or increasing the number of references from other authors.

Comment 2

Section 1 (line 58) + Reference list

Reference [6] is missing in the reference list.

Comment 3

Section 2

This section lacks of additional information for the readers and should be complemented. For instance:

- photos of: used fine RA, samples of the obtained RC, testing procedures, …;

- mix designs of concretes, including granulometric curves of the used aggregates;

- etc.

Comment 4

Some aspects of the organization of the manuscript must be revised. For instance, Fig. 1 and 2 are referred and discussed in page 4, but they appear only in pages 10 to 12.

Comment 5

Section 3, Line 225

I think it should be “exceeding 15%”. Please check!

Comment 6

Section 5

This section is more like a summary or abstract of the manuscript, and not a conclusion section. It should be entirely revised.

Comment 7

Section 5

The application of the proposed technical specification is somewhat limited since it was based on an experimental campaign using fine RA from three recycling plants in Portugal. How this proposal can be extended to other countries? This aspect should be briefly discussed since the manuscript aims to be published in an international journal.

Author Response

Reviewer #3 COMMENTS/CORRECTIONS:

Comment 1: The submitted manuscript presents a proposal for a technical specification for using fine recycled aggregates (RA) from construction and demolition waste (CDW) in concrete production. For this, several experimental results on recycled concrete (RC) obtained by the authors and produced using fine RA from three recycling plants in Portugal, with different contents of fine RA as replacement of fine natural aggregates, were analysed. The structure of the proposed technical specification was inspired from an existing Portuguese technical specification for using coarse RA from CDW in concrete production, in which fine RA were excluded. The manuscript summarizes the experimental campaign and discusses the results. It Is shown that for contents of fine RA not to high, the decrease on the properties and durability of the obtained RC are not excessively reduced. The proposal for the technical specification is presented, in order to provide recommendations and establish minimum requirements for the use of fine RA in the production of RC. Finally, the use of the proposed technical specification is exemplified for RC using fine RA from the CDW studied in the experimental campaign. The topic that is developed in the study is interesting and important since it is related with the sustainability of construction sector, and deals with fine RA in concrete for which existent legislation is still practically inexistent. The proposed technical specification could help task groups to propose future recommendations and also industries from the construction sector. I made some suggestions in order to improve the manuscript. The authors should take the suggestions into account and revise their manuscript.

Reply: The authors gratefully acknowledge the general comment made by Reviewer #3.

Action: The authors considered the suggestions made in the following comments.

Comment 2: Section 1. In the literature review, the number of cited articles from at least one of the authors (self-citations) is to high (about 50%). Given the high number of published studies on the topic that can be found in the literature, this issue should be revised by reducing the number of self-citations and/or increasing the number of references from other authors.

Reply: The authors agree with this comment made by Reviewer #3 and, therefore, increased the number of studies on the topic by other authors cited and decreased those configuring self-citations.

Action: References [14], [18], [21] and [24] were replaced in the reference list.

Comment 3: Section 1 (line 58) + Reference list. Reference [6] is missing in the reference list.

Reply: The authors thank Reviewer #3 for this comment.

Action: Reference [6] was added to the reference list.

Comment 4: Section 2. This section lacks of additional information for the readers and should be complemented. For instance:

- photos of: used fine RA, samples of the obtained RC, testing procedures, …;

- mix designs of concretes, including granulometric curves of the used aggregates;

- etc.

Reply: The authors agree to some extent with this comment made by Reviewer #3. Even though the scope of the work is not the experimental work (readers are encouraged to read references [10] and [23] for detailed insight), the authors recognize that some additional elements can be added, to increase the self-sufficiency of section 2.

Action: The following additions/changes were made, as suggested:

  • The Faury curve was referred in the scope of the RC mix composition and the fact that aggregates were replaced sieve by sieve was also addressed (lines 106-108);
  • The mixing procedure was briefly described (lines 113-115);
  • The number of specimens used in each test was given (as also suggested by Reviewer #4, in comment 4) (lines 126-128);
  • Illustrative images of Europontal’s FRA and of the shrinkage test setup were included in (new) Figure 1 (line 129);
  • The remittance of readers for detailed descriptions of experimental campaigns and obtained results was placed at the beginning of the section, rather than at the end, to enforce this idea (lines 101-102).

Comment 5: Some aspects of the organization of the manuscript must be revised. For instance, Fig. 1 and 2 are referred and discussed in page 4, but they appear only in pages 10 to 12.

Reply: The authors understand this comment made by Reviewer #3 in the sense that explicitly addressing Figures 1 and 2 (now numbered as Figures 2 and 3) in page 4 and only presenting them in pages 10 to 12 may suggest some lack of organization. In fact the word ‘ahead’ was intentionally used to express such ‘distance’. Indeed the purpose of Figures 1 and 2 (now numbered as Figures 2 and 3) is not that of showing the experimental results (this is made in Tables 1 and 2) but to frame the variation of properties into the defined concrete classes and corresponding percentage decreases. However, the authors believe they can additionally aid readers in the interpretation of results shown in Tables 1 and 2, thus referring them.

Action: Figures 1 and 2 (now numbered as Figures 2 and 3) in section 2/page 4 (lines 148-149) are now referred inside brackets to decrease the emphasis of the allusion and to make it consist more of a mere suggestion to readers.

Comment 6: Section 3, Line 225. I think it should be “exceeding 15%”. Please check!

Reply: The authors thank Reviewer #3 for this comment. The percentage quoted was in fact incorrect.

Action: The percentages was corrected to ‘15%’ in line 225.

Comment 7: Section 5. This section is more like a summary or abstract of the manuscript, and not a conclusion section. It should be entirely revised.

Reply: The authors agree with this comment made by Reviewer #3 as also stated in the reply to (similar) comment 5 made by Reviewer #1: conclusions were too narrative.

Action: The authors modified the conclusions accordingly.

Comment 8: Section 5. The application of the proposed technical specification is somewhat limited since it was based on an experimental campaign using fine RA from three recycling plants in Portugal. How this proposal can be extended to other countries? This aspect should be briefly discussed since the manuscript aims to be published in an international journal.

Reply: The authors thank Reviewer #3 for this comment. Indeed this aspect was not very clear in the manuscript. Despite the fact that the aggregates are from Portuguese recycling plants, as mentioned, it can be seen that their classing in the specification proposal is made based on their composition rather than on their geographical origin, as per Table 3. Hence, the main idea is that similar compositions of aggregates (irrespective of the country considered) will result in concrete with similar properties, thus giving the proposed specification a broader scope than that of the Portuguese territory.

Action: The clarification to the aspect raised by Reviewer #3 was included in the abstract (lines 27-29), introduction (lines 96-98) and conclusions of manuscript (lines 349-351).

Reviewer 4 Report

1) Very practical and valuable paper refers to assessment of fine recycled aggregates from CDW. 

2) The paper is consistent and well written.

3) Please add short objectives description (maybe in points) at the end of Introduction paragraph. Not describe the content of the paper.

4) Please correct the Figures (1 and 2). Description presented in Figs is not clearly visible (Class 1 - Class 4).

5) How many specimens vere tested for each value presented in Fig. 1 and 2?

6) Please consider to add the statistical parameter COV to the results presented in Figs. 

Author Response

Reviewer #4 COMMENTS/CORRECTIONS:

Comment 1: 1) Very practical and valuable paper refers to assessment of fine recycled aggregates from CDW. 2) The paper is consistent and well written.

Reply: The authors gratefully acknowledge the general comments made by Reviewer #4.

Action: The authors considered the suggestions made in the following comments.

Comment 2: Please add short objectives description (maybe in points) at the end of Introduction paragraph. Not describe the content of the paper.

Reply: The authors agree with this comment made by Reviewer #4 in the sense that the objectives of the work should be emphasized at the end of the introduction.

Action: The authors added the objectives of the work (in points/bullets) at the end of the introduction (lines 94-98).

Comment 3: Please correct the Figures (1 and 2). Description presented in Figs is not clearly visible (Class 1 - Class 4).

Reply: The authors thank Reviewer #4 for this comment. Due to some formatting issue indeed ‘Class 1’ to ‘Class 4’ descriptions were not clearly visible in Figures 1 and 2 (now numbered as Figures 2 and 3).

Action: The authors corrected the Figures 1 and 2 (now numbered as Figures 2 and 3) accordingly.

Comment 4: How many specimens were tested for each value presented in Fig. 1 and 2?

Reply: The authors thank Reviewer #4 for this comment. Indeed, the reference to the number of specimens used to obtain each property evaluated was missing. 3 specimens were used, in general, in each test with exception of the water absorption by immersion and compressive strength in cubes in which, respectively, 4 and 5 specimens were used.

Action: This information was added to the text (lines 126-128).

Comment 5: Please consider to add the statistical parameter COV to the results presented in Figs. 2

Reply: The authors thank Reviewer #4 for the suggestion made in this comment and agree that such information would contribute to a more representative display of the results. In fact, similarly to what was performed in Tables 1 and 2, also the standard deviation was attempted do be included in Figures 1 and 2 (now numbered as Figures 2 and 3). However, it was found that, due to the high amount of information shown, these figures lost readability. Since the standard deviation is already presented in Tables 1 and 2, the authors believe that statistical significance is, nevertheless, ensured in this way.

Action: Given the aforementioned, the authors would like to maintain Figures 1 and 2 (now numbered as Figures 2 and 3) as they are. Nevertheless, they added the following (in line 292) to enforce that only the average values are being presented: ‘(of the average values)’.

Round 2

Reviewer 3 Report

I´m satisfied with the authors’ replies to my comments and I also consider that all my suggestions and concerns have been properly explained and considered by the authors to improve the article.

Author Response

INTRODUCTION

The authors would like to thank the Editor and Reviewers for the careful examination of the revised paper and valuable comments. Next, the comments made by the Editor and Reviewer #3 are addressed: (i) the Editor/Reviewer comment/question is presented first (in italic), (ii) the authors reply is given next and then (iii) the action prompted by the reply is described. It is worth noting that the “Revised manuscript” contains all the modifications incorporated. According to the Editor’s and Reviewers’ suggestions, the materials added and corrections made are highlighted in blue.

Reviewer #3 COMMENTS/CORRECTIONS:

Comment 1: I´m satisfied with the authors’ replies to my comments and I also consider that all my suggestions and concerns have been properly explained and considered by the authors to improve the article.

Reply: The authors gratefully acknowledge the comment made by Reviewer #3.

Action: No actions are required.